# Ketamine Improves Desensitization of µ-Opioid Receptors Induced by Repeated Treatment with Fentanyl but Not with Morphine

**DOI:** 10.3390/biom12030426

**Published:** 2022-03-10

**Authors:** Yusuke Mizobuchi, Kanako Miyano, Sei Manabe, Eiko Uezono, Akane Komatsu, Yui Kuroda, Miki Nonaka, Yoshikazu Matsuoka, Tetsufumi Sato, Yasuhito Uezono, Hiroshi Morimatsu

**Affiliations:** 1Department of Anesthesiology and Resuscitology, Okayama University Graduate School of Medicine, Dentistry and Pharmaceutical Sciences, 2-5-1 Shikatacho, Kita-ku, Okayama-shi 700-8558, Japan; pkb37iph@s.okayama-u.ac.jp (Y.M.); pb9b45wr@okayama-u.ac.jp (H.M.); 2Department of Pain Control Research, The Jikei University School of Medicine, 3-25-8 Nishi-Shimbashi, Minato-ku, Tokyo 105-8461, Japan; k.miyano@jikei.ac.jp (K.M.); eiyu0825@gmail.com (E.U.); minonaka@jikei.ac.jp (M.N.); 3Department of Anesthesiology and Critical Care Medicine, National Cancer Center Hospital, 5-1-1 Tsukiji, Chuo-ku, Tokyo 104-0045, Japan; tesatoh@ncc.go.jp; 4Department of Anesthesiology and Resuscitology, Okayama University Hospital, 2-5-1 Shikatacho, Kita-ku, Okayama-shi 700-8558, Japan; me421081@s.okayama-u.ac.jp (S.M.); matsuoka2@okayama-u.ac.jp (Y.M.); 5Department of Anesthesiology and Pain Medicine, Juntendo University Graduate School of Medicine, 2-1-1 Hongo, Bunkyo-ku, Tokyo 113-8421, Japan; a-komats@juntendo.ac.jp (A.K.); ykuroda701@gmail.com (Y.K.); 6Supportive and Palliative Care Research Support Office, National Cancer Center Hospital East, 6-5-1 Kashiwanoha, Kashiwa-shi 277-8577, Japan

**Keywords:** µ-opioid receptor, desensitization, tolerance, fentanyl, morphine, ketamine, G protein receptor kinase, β-arrestin

## Abstract

The issue of tolerance to continuous or repeated administration of opioids should be addressed. The ability of ketamine to improve opioid tolerance has been reported in clinical studies, and its mechanism of tolerance may involve improved desensitization of μ-opioid receptors (MORs). We measured changes in MOR activity and intracellular signaling induced by repeated fentanyl and morphine administration and investigated the effects of ketamine on these changes with human embryonic kidney 293 cells expressing MOR using the CellKey™, cADDis cyclic adenosine monophosphate, and PathHunter^®^ β-arrestin recruitment assays. Repeated administration of fentanyl or morphine suppressed the second MOR responses. Administration of ketamine before a second application of opioids within clinical concentrations improved acute desensitization and enhanced β-arrestin recruitment elicited by fentanyl but not by morphine. The effects of ketamine on fentanyl were suppressed by co-treatment with an inhibitor of G-protein-coupled receptor kinase (GRK). Ketamine may potentially reduce fentanyl tolerance but not that of morphine through modulation of GRK-mediated pathways, possibly changing the conformational changes of β-arrestin to MOR.

## 1. Introduction

Opioids have been used for the relief of cancer [1], perioperative [2], and critical-illness-related [3] pain, but increase in usage due to tolerance is an issue that should be addressed [4,5,6]. Tolerance is defined as a reduction in drug efficacy due to prolonged or repeated administration, leading to reduced drug effects and increased dosage to maintain the analgesic effects. These dosage increases may accelerate the appearance of side effects, including respiratory depression, constipation, and addiction [7]. Opioid tolerance could be caused by signaling desensitization, receptor downregulation, upregulation of drug metabolism, and initiation of compensatory/opponent processes [8,9]. Therefore, elucidating the mechanism of opioid tolerance is important to develop tolerance-prevention strategies and novel clinical treatments.

Opioid receptors (ORs) belong to the G-protein-coupled receptor (GPCR) family and are classified into several subtypes. The major subtypes include μ-(MOR), δ-(DOR), κ-(KOR), and nociceptin (NOR), whereas opioid analgesics are mainly mediated by MOR [10,11]. When an agonist ligand binds to the OR, two major intracellular signaling pathways are activated: the G protein-mediated pathway and the β-arrestin-mediated pathway [12]. The former activates the G protein and induces a decrease in intracellular cyclic adenosine monophosphate (cAMP) levels through the inhibition of adenylate cyclase, which is associated with analgesia. The latter is activated by phosphorylation of the carboxyl terminus of ORs via the G-protein-coupled receptor kinase (GRK), and β-arrestin binds to the phosphorylated sites, inducing internalization of ORs via endocytosis and subsequent intracellular signaling or degradation of ORs by lysosomes [13]. After endocytosis, the resensitized receptors recycle back to the cell membrane by vesicular delivery for subsequent activation [14]. A previous study showed reduced constipation and respiratory depression, presumably due to decreased receptor desensitization, in β-arrestin-2-knockout mice [15]. Thereafter, the cellular response of the β-arrestin-mediated pathway via ORs has been believed to be primarily associated with side effects. However, recent studies have failed to replicate such findings [16]; thus, the debate remains open [17].

The phenomenon where intracellular signals are reduced by sustained or repeated receptor stimulation is known as receptor desensitization [18]. MOR desensitization has been shown to be mediated by phosphorylation of the agonist-stimulated receptor by GRK2 followed by binding of β-arrestin to the phosphorylated receptors [19]. Desensitization attributed to continuous MOR activation may be involved in the mechanism of tolerance, but this has not been determined [8].

Ketamine is a phenylcyclohexylamine derivative and a dissociative anesthetic with clinical use since 1970. In addition to its anesthetic effect, ketamine exerts analgesic and anti-inflammatory effects and an antidepressant activity [20]. Despite having side effects, such as dissociation and psychological symptoms, ketamine remains in use as an anesthetic, analgesic, and antidepressant. Previous studies have reported the efficacy of using ketamine in patients with opioid tolerance and inadequate analgesia in clinical settings [21,22,23]. Ketamine is a known N-methyl-D-aspartate (NMDA) receptor antagonist, but its effects on ORs have also been reported [24]. The combination of ketamine with opioids enhances phosphorylation of ERK1/2 in MOR. Although ketamine modulates MOR signaling, the mechanism behind this modulation (including whether it acts at the receptor or downstream signaling) and its effect on receptor desensitization remain to be clarified.

Accordingly, in this study, we evaluated the changes in MOR activity and intracellular signaling following repeated administration of fentanyl and morphine using human embryonic kidney 293 (HEK293) cells expressing MOR. In addition, we focused on the effects of ketamine administration on acute desensitization induced by repeated opioid administration.

## 2. Materials and Methods

### 2.1. Chemicals

The following reagents were used: fentanyl citrate injection solution (Janssen Pharmaceutical K.K., Tokyo, Japan), morphine hydrochloride (Takeda Pharmaceutical, Tokyo, Japan), ketamine hydrochloride (Sigma-Aldrich, Saint Louis, MO, USA), (+)-MK-801 hydrogen (Sigma-Aldrich), forskolin (FUJIFILM Wako Pure Chemical Corporation, Osaka, Japan), CMPD101 (MedChemExpress, Monmouth Junction, NJ, USA), U0126 (Promega, Madison, WI, USA), c-Jun N-terminal kinase (JNK) inhibitor II (Sigma-Aldrich), and Ro 31-8220 (MedChemExpress). Fentanyl, morphine, and ketamine were diluted with H_2_O, while the other reagents were diluted with dimethyl sulfoxide.

### 2.2. Construction of Plasmids and Generation of Stable Cell Lines

The process of plasmid construction and generation of stable cell lines for MORs has been described previously [25]. Halotag^®^ fused MOR (Halotag^®^ MOR, Kazusa DNA Research Institute, Chiba, Japan) and the pGlosensor™-22F plasmid (pGS22F, Promega) were amplified according to the manufacturer’s instructions. HEK293 cells (ATCC^®^, Manassas, VA, USA) stably expressing both Halotag^®^ MOR and pGS22F were generated by transfection of the constructed plasmids using the Lipofectamine reagent (Life Technologies, Carlsbad, CA, USA). These were selected based on OR activity measured using the CellKey™ assay or the cADDis^®^ cAMP assay.

### 2.3. Cell Culture

HEK293 cells stably expressing Halotag^®^ MOR/pGS22F were cultured in Dulbecco’s modified Eagle medium supplemented with 10% fetal bovine serum, 1% penicillin/streptomycin, 5 μg/mL puromycin (InvivoGen, San Diego, CA, USA), and 100 μg/mL hygromycin (FUJIFILM Wako Pure Chemical Corporation) in a humidified atmosphere containing 95% air and 5% CO_2_ at 37 °C.

### 2.4. CellKey™ Assay

The procedures in the present study were performed following a protocol described previously [25]. The CellKey™ assay system, a label-free, cell-based assay for detecting GPCR activity, has also been described previously [26]. Briefly, cells stably expressing Halotag^®^ MOR/pGS22F were seeded at densities of 4.0 × 10^4^ in poly-D-Lysine (Sigma Aldrich)-coated CellKey™ 96-well microplates and incubated for 24 h. The medium was replaced with a CellKey™ buffer composed of Hank’s balanced salt solution (in mM: 1.3 CaCl_2_·2H_2_O, 0.81 MgSO_4_, 5.4 KCl, 0.44 KH_2_PO_4_, 4.2 NaHCO_3_, 136.9 NaCl, 0.34 Na_2_HPO_4_ and 5.6 D-glucose) containing 20 mM 4-(2-hydroxyethyl)-1-piperazineethanesulfonic acid and 0.1% bovine serum albumin. Repeated administration of the same doses of fentanyl or morphine was performed as follows. (1) Cells were incubated at 28 °C for 30 min; (2) changes in the impedance current (ΔZiec) in each well were measured at 10 s intervals for up to 30 min, with the first 5 min as the baseline, and ΔZiec measurements were obtained for 25 min after administration of each opioid (first treatment); (3) the cells were incubated at 28 °C for 30 min after washing; (4) ΔZiec were measured and treated with the same dose of each opioid (second treatment), same as for the first treatment. Ketamine, MK-801, and other inhibitors were administrated 30 min before the first or second treatments, respectively. The ΔZiec values for each sample were normalized using the values of the negative control sample.

### 2.5. cADDis cAMP Assay

The cADDis cAMP assay system using the cADDis cAMP assay kit (#U0200G) (Montana Molecular, Bozeman, MT, USA) has been described previously [27]. Briefly, cells were seeded at 5.0 × 10^4^ cells/well (Halotag^®^ MOR/pGS22F) on black-walled, clear, flat-bottomed 96-well plates with recombinant BacMam virus expressing the cADDis sensor and 0.6 µM sodium butyrate and incubated for 24 h at 5% CO_2_ at 37 °C. The medium was replaced with 100 µL of Krebs solution, and the cells were incubated at 28 °C for 30 min in the dark. The cells were stimulated with the indicated opioids (first treatment) for 30 min after incubation. The wells were washed with 100 µL Krebs solution, and the cells were incubated again at 28 °C for 30 min in the dark before the measurement of the second stimulation (second treatment). Ketamine, MK-801, and other inhibitors were administrated 30 min before each opioid stimulation as was performed for the CellKey™ assay. Cell fluorescence was measured from the plate bottom using excitation/emission wavelengths of 485 and 525 nm, respectively, using the FlexStation 3 (Molecular Devices, LLC., San Jose, CA, USA). Changes in fluorescence in each well were measured at 26 s intervals for up to 30 min while considering the first 5 min as the baseline, and the cells were stimulated with 50 µM forskolin to increase the cAMP levels for 25 min. After the signal plateaued, cells were stimulated with the second opioid administration, and fluorescence changes in each well were measured for 60 min. Data were transformed to change in fluorescence over the initial fluorescence (ΔF/F_0_).

### 2.6. PathHunter^®^ eXpress β-Arrestin Assay

The β-arrestin recruitment assays have been described previously [28] and were performed according to the protocol for PathHunter^®^ (DiscoverX, Fremont, CA, USA). U2OS OPRM1 cells were seeded at a density of 1.0 × 10^4^ cells/well in 96-well clear-bottom white plates and incubated for 48 h at 5% CO_2_ at 37 °C. The medium was replaced with 100 µL of cell plating reagent, and the cells were treated with each opioid and incubated at 28 °C for 30 min in the dark. After washing the wells with 100 µL cell plating reagent, the cells were incubated again at 28 °C for 30 min in the dark before the measurement of the second stimulation. Ketamine, MK-801 and other inhibitors were administrated 30 min before each opioid stimulation as was performed for the CellKey™ assay. Luminescence intensities were measured from the plate bottom using excitation/emission wavelengths of 485 and 525 nm, respectively, using the FlexStation 3 (Molecular Devices). The cells were stimulated for 90 min with the second opioid administration at 37 °C and 5% CO_2_. After PathHunter^®^ working detection solution was added, luminescence changes in each well were measured every 26 s for 60 min. Data are expressed as the amount of relative light units.

### 2.7. Statistical Analysis

Data analyses were performed using GraphPad Prism 9 (GraphPad Software, La Jolla, CA, USA). Data are presented as means with standard error of the mean (SEM) for at least three independent experiments. Statistical analysis was performed using the one-way or two-way analysis of variance (ANOVA) followed by the post hoc Tukey’s multiple comparisons test (GraphPad Prism 9). A *p* < 0.05 was considered statistically significant.

## 3. Results

### 3.1. Effects of Ketamine on Decrease in MOR Activity Induced by Repeated Opioid Administration Using the CellKey™ Assay

#### 3.1.1. Repeated Administration of Fentanyl or Morphine Decreased MOR Activity

We evaluated the changes in MOR activity with repeated administration of the same doses of fentanyl and morphine with the CellKey™ system, which can detect GPCR activity as change in cellular impedance [26]. HEK293 cells expressing Halotag^®^ MOR/pGS22F were treated with fentanyl or morphine (first administration) for 25 min. After washing and incubation for 30 min, the same dose of each opioid was administered (second administration) and cellular impedance was measured (Figure 1a). A two-way ANOVA revealed significant effects of dose (fentanyl: F (4, 62) = 425.1, *p* < 0.0001, η_p_^2^ = 0.965; morphine: F (4, 62) = 454.4, *p* < 0.0001, η_p_^2^ = 0.967), number of doses (fentanyl: F (1, 62) = 710.4, *p* < 0.0001, η_p_^2^ = 0.920; morphine: F (1, 62) = 33.1, *p* < 0.0001, η_p_^2^ = 0.348), and interaction (fentanyl: F (4, 62) = 179.1, *p* < 0.0001, η_p_^2^ = 0.920; morphine: F (4, 62) = 12.5, *p* < 0.0001, η_p_^2^ = 0.447). A post hoc Tukey’s test showed that, compared with treatment with vehicle to fentanyl, repeated administration of fentanyl to fentanyl (1–1000 nM) at the same dose decreased MOR activity in a dose-dependent manner (Figure 1b). In contrast, repeated administration with a high dose of morphine (10,000 nM) decreased MOR activity (Figure 1c).

#### 3.1.2. Treatment with Ketamine before the Second Administration of Fentanyl Recovered the Decrease in MOR Activity

To evaluate the effects of ketamine on the second administration of fentanyl or morphine, we first examined changes in pretreatment with ketamine on single administration (first administration) of these opioids. Ketamine was administered for 30 min before a single administration of fentanyl or morphine (Figure 2a). A two-way ANOVA revealed a significant effect of dose (fentanyl: F (4, 74) = 463.8, *p* < 0.0001, η_p_^2^ = 0.962; morphine: F (4, 58) = 1568, *p* < 0.0001, η_p_^2^ = 0.991), but no significant effects of ketamine 100 µM pretreatment (fentanyl: F (1, 74) = 0.028, *p* = 0.868, η_p_^2^ < 0.001; morphine: F (1, 58) = 3.34, *p* = 0.073, η_p_^2^ = 0.054) or interaction (fentanyl: F (4, 74) = 0.037, *p* = 0.997, η_p_^2^ = 0.002; morphine: F (4, 58) = 0.782, *p* = 0.541, η_p_^2^ = 0.051). A post hoc Tukey’s test showed that ketamine did not affect the response induced by fentanyl or morphine even at a high ketamine dose (100 µM) (Figure 2b,c). The results of the two-way ANOVA followed by the post hoc Tukey’s test for the fentanyl by ketamine dose are available in Appendix A.

We next measured changes in pretreatment with ketamine (1–100 µM) on the second administration of fentanyl and morphine (Figure 2d). A one-way ANOVA revealed significant effects of combinations of drugs on change in impedance (Figure 2e: F (7, 46) = 44.8, *p* < 0.0001, η^2^ = 0.872; Figure 2f: F (7, 46) = 36.1, *p* < 0.0001, η^2^ = 0.846; Figure 2g: F (7, 46) = 281.8, *p* < 0.0001, η^2^ = 0.977; Figure 2h: F (7, 46) = 99.2, *p* < 0.0001, η^2^ = 0.938). A post hoc Tukey’s test showed that ketamine at doses higher than 30 µM improved the decrease in MOR activity caused by the second fentanyl (10–100 nM) application (Figure 2e,f), but not in that caused by 1000 nM fentanyl (Figure 2g). In contrast, ketamine did not recover the decrease in MOR activity induced by repeated administration of morphine (Figure 2h).

#### 3.1.3. Mechanisms of Ketamine Pretreatment on the Decrease in MOR Activity Caused by the Second Fentanyl Administration

To confirm whether the action of ketamine was attributable to the inhibition of the NMDA receptor activity, we examined the effects of MK-801, the uncompetitive antagonist of the NMDA receptor, on the second administration of fentanyl. MK-801 (1–100 µM) was administered for 30 min before the second administration of 100 nM fentanyl (Figure 3a). A one-way ANOVA revealed a significant effect of combinations of drugs on change in impedance (F (8, 51) = 159.5, *p* < 0.0001, η^2^ = 0.962). A post hoc Tukey’s test showed that MK-801 failed to inhibit the decrease in MOR activity induced by the second fentanyl administration (Figure 3b).

We investigated the effects of several intracellular signal inhibitors (CMPD101 (a GRK 2,3 inhibitor), U0126 (a mitogen-activated protein kinase (MEK) inhibitor), JNK inhibitor II, and Ro31-8220 (a protein kinase C (PKC) inhibitor)) on the ketamine-induced improvement of the decrease in MOR activity evoked by fentanyl. Each inhibitor was administered concurrently with ketamine (Figure 3c). A one-way ANOVA revealed significant effects of combinations of drugs on change in impedance (Figure 3d: F (8, 51) = 43.2, *p* < 0.0001, η^2^ = 0.871; Figure 3e: F (8, 51) = 24.1, *p* < 0.0001, η^2^ = 0.791; Figure 3f: F (8, 51) = 36.5, *p* < 0.0001, η^2^ = 0.851; Figure 3g: F (8, 51) = 67.1, *p* < 0.0001, η^2^ = 0.913). A post hoc Tukey’s test showed that only CMPD101 significantly cancelled the ketamine-induced improvement of the decrease in MOR activity evoked by repeated administration of 100 nM fentanyl (Figure 3d–g). No treatment with inhibitors in the absence of ketamine affected the decrease in MOR activity evoked by repeated administration of fentanyl (Appendix A).

### 3.2. Effects of Ketamine on the Decrease in Intracellular cAMP Induced by the Second Opioid Administration with the cADDis cAMP Assay

#### 3.2.1. Repeated Administration of Fentanyl or Morphine Suppressed the Decrease in Intracellular cAMP

The cAMP assay with the cADDis sensor was performed to detect the activity of the Gi/o protein. The cADDis sensor used in this study increases fluorescence intensity when the levels of intracellular cAMP decrease in response to the activation of Gi/o protein. Conversely, the cADDis sensor decreases fluorescence intensity when the level of intracellular cAMP increases. A two-way ANOVA revealed significant effects of dose (fentanyl: F (4, 50) = 38.4, *p* < 0.0001, η_p_^2^ = 0.754; morphine: F (4, 50) = 50.9, *p* < 0.0001, η_p_^2^ = 0.533), number of doses (fentanyl: F (1, 50) = 90.2, *p* < 0.0001, η_p_^2^ = 0.643; morphine: F (1, 50) = 22.2, *p* < 0.0001, η_p_^2^ = 0.223), and interaction (fentanyl: F (4, 50) = 13.9, *p* < 0.0001, η_p_^2^ = 0.526; morphine: F (4, 50) = 5.14, *p* = 0.002, η_p_^2^ = 0.225). A post hoc Tukey’s test showed that, compared with treatment with vehicle to fentanyl, the second administration of fentanyl (10–1000 nM) at the same dose suppressed the decrease in intracellular cAMP in a dose-dependent manner (Figure 4a). In contrast, only repeated administration with a high dose of morphine (10,000 nM) suppressed the decrease in intracellular cAMP (Figure 4b).

#### 3.2.2. Pretreatment with Ketamine before the Second Administration of Fentanyl Recouped the Rescue of Intracellular cAMP Induced by the Second Fentanyl Administration

We measured the effects of ketamine on repeated administration of fentanyl and morphine. Ketamine (10–100 µM) was administered for 30 min before the second administration of fentanyl and morphine. A one-way ANOVA revealed significant effects of combinations of drugs on ΔF/F_0_ (Figure 4c: F (5, 30) = 11.1, *p* < 0.0001, η^2^ = 0.650; Figure 4d: F (5, 30) = 61.8, *p* < 0.0001, η^2^ = 0.912; Figure 4e: F (5, 30) = 12.9, *p* < 0.0001, η^2^ = 0.683). A post hoc Tukey’s test showed that ketamine at doses higher than 30 µM recovered the rescue of intracellular cAMP caused by the repeated fentanyl (10–100 nM) administration (Figure 4c,d). Ketamine did not recover the rescue of intracellular cAMP caused by repeated morphine administration (Figure 4e).

#### 3.2.3. Mechanisms of Ketamine on the Rescue of Intracellular cAMP Caused by Repeated Fentanyl Administration

In the CellKey™ assay, CMPD101 cancelled the ketamine-induced improvement in the rescue of intracellular cAMP induced by repeated fentanyl administration. U0126 tends to suppress the effect of ketamine but not to a great extent. Therefore, we investigated the effects of these inhibitors on the ketamine-induced improvement in the rescue of intracellular cAMP caused by repeated fentanyl administration. Each inhibitor was administered concurrently with ketamine. A one-way ANOVA revealed significant effects of combinations of drugs on ΔF/F_0_ (Figure 5a: F (8, 45) = 40.3, *p* < 0.0001, η^2^ = 0.877; Figure 5b: F (8, 49) = 29.3, *p* < 0.0001, η^2^ = 0.827). A post hoc Tukey’s test showed that CMPD101 (0.01–10 µM) did not improve the rescue of intracellular cAMP caused by repeated administration of 100 nM fentanyl (Appendix A). However, CMPD101 (1–10 µM) significantly cancelled the ketamine-induced improvement in rescue of intracellular cAMP caused by repeated administration of 100 nM fentanyl (Figure 5a). U0126 (0.01–10 µM) did not affect the rescue of intracellular cAMP caused by repeated administration of 100 nM fentanyl (Appendix A) and did not affect the ketamine-induced improvement in the rescue of intracellular cAMP caused by repeated administration of 100 nM fentanyl (Figure 5b).

### 3.3. Effects of Ketamine on Recruitment of β-Arrestin to MOR Induced by Repeated Administration of Opioids Using the PathHunter^®^ eXpress β-Arrestin Assay

#### 3.3.1. Effect of Treatment with Ketamine on the Enhanced β-Arrestin Recruitment to MOR Induced by Repeated Administration of Fentanyl

We performed the PathHunter^®^ eXpress β-arrestin assay to analyze the action of ketamine on the β-arrestin-mediated pathway. Ketamine (10–100 µM) was administered for 30 min before the second administration of fentanyl or morphine as was performed for the CellKey™ and cADDis cAMP assays. A one-way ANOVA revealed significant effects of combinations of drugs on amount of luminescence (Figure 6a: F (5, 30) = 12.9, *p* < 0.0001, η^2^ = 0.683; Figure 6b: F (5, 30) = 40.0, *p* < 0.0001, η^2^ = 0.870; Figure 6c: F (5, 30) = 125.9, *p* < 0.0001, η^2^ = 0.955). A post hoc Tukey’s test showed that ketamine at doses higher than 30 µM enhanced the level of β-arrestin recruitment for MOR induced by the second fentanyl administration (10 and 100 nM) (Figure 6a,b). Ketamine failed to enhance the level of β-arrestin recruitment for MOR induced by the repeated morphine administration (Figure 6c).

#### 3.3.2. Mechanisms of Ketamine on the Enhancement of β-Arrestin Recruitment to MOR Induced by Repeated Administration of Fentanyl

We investigated the effects of CMPD101 and U0126 on the ketamine-induced enhancement of β-arrestin recruitment to MOR induced by repeated fentanyl administration as was performed for the CellKey™ and cADDis cAMP assays. Each inhibitor was administered concurrently with ketamine. A one-way ANOVA revealed significant effects of combinations of drugs on amount of luminescence (Figure 7a: F (7, 40) = 70.1, *p* < 0.0001, η^2^ = 0.925; Figure 7b: F (7, 40) = 45.2, *p* < 0.0001, η^2^ = 0.888). A post hoc Tukey’s test showed that 10 µM CMPD101 inhibited the level of β-arrestin recruitment to MOR induced by the repeated administration of 100 nM fentanyl (Appendix A). In addition, CMPD101 (1–10 µM) significantly cancelled the ketamine-induced enhancement of β-arrestin recruitment to MOR induced by the repeated administration of 100 nM fentanyl (Figure 7a). U0126 (0.01–10 µM) did not affect the level of β-arrestin recruitment to MOR induced by the repeated administration of 100 nM fentanyl (Appendix A) and did not affect ketamine-induced enhancement of β-arrestin recruitment to MOR induced by the repeated administration of 100 nM fentanyl (Figure 7b).

## 4. Discussion

In the present study, we established an assay system using CellKey™ to evaluate acute MOR desensitization. Repeated administration of the same dose of fentanyl (10, 100, 1000 nM) and morphine (10,000 nM) at 60 min intervals resulted in a decrease in MOR activity compared with single administration. We did not increase the concentration of morphine considering that the fentanyl is 100 times more potent than morphine [25]. Because repeated administration of fentanyl and morphine suppressed MOR activity at the same dose in the CellKey™ assay, we used this assay as a model for acute MOR desensitization. Treatment with ketamine before the second administration of fentanyl or morphine recovered the decrease in MOR activity induced by fentanyl but not that induced by morphine. Several intracellular signal molecules, such as GRK, MEK, JNK, and PKC, have been found to be associated with MOR desensitization [29]. During simultaneous treatment of intracellular signaling inhibitors with ketamine, only CMPD101, an inhibitor of GRKs, significantly cancelled the ketamine-induced improvement of decrease in MOR activity. In the cADDis cAMP assay, repeated administration of fentanyl or morphine suppressed the decrease in intracellular cAMP similar to the results of the CellKey™ assay. Treatment with ketamine before the second administration of fentanyl, but not of morphine, recovered the rescue of intracellular cAMP. The ketamine-induced improvement in the rescue of intracellular cAMP caused by repeated administration of fentanyl was cancelled by co-treatment with CMPD101 but not by co-treatment with U0126, an inhibitor of MEK1/2. Finally, our PathHunter^®^ eXpress β-arrestin assay showed that ketamine at doses higher than 30 µM enhanced the level of β-arrestin recruitment for MOR induced by repeated fentanyl, but not by repeated morphine administration. The ketamine-induced enhancement of β-arrestin recruitment to MOR caused by repeated fentanyl administration was cancelled by co-treatment with CMPD101 but not with U0126.

Ketamine has recently attracted attention as a treatment for depression, and analysis of its mechanism of action and affinity for receptors is underway [30]. Ketamine is known to be an NMDA-type glutamate receptor antagonist [31], but it has also been reported to act directly on α-amino-3-hydroxy-5-methyl-4-isoxazole propionic acid (AMPA) receptors [20], orexin-1 receptors [32], and ORs [33,34]. However, our present study showed that ketamine did not directly activate MOR using the CellKey™ assay even at a higher dose (100 μM). We found that ketamine, but not MK-801, improved fentanyl desensitization, suggesting that the improvement in opioid desensitization induced by ketamine affects MOR but not via the NMDA receptors.

Our present study indicated that ketamine improved the desensitization of MOR induced by fentanyl, but not that by morphine, suggesting that desensitization induced by fentanyl and morphine might occur according to different mechanisms. It has been reported that, of the GRK subtypes, fentanyl mainly activates GRK2/3, whereas morphine activates GRK5 [29]. Both GRK2/3 and GRK5 have also been shown to be associated with desensitization of GPCR, but the mechanism may differ for each subtype [19,35,36]. Fentanyl has a strong effect on β-arrestin recruitment via GRK phosphorylation, which induces desensitization, whereas morphine has a weak effect on β-arrestin recruitment, and PKC is involved in the process [37]. The reason is uncertain at present, but it may be possible that the action mechanisms of ketamine are related to the phosphorylation site of MORs by GRK2/3, but not by GRK5, and the subsequent recruitment of β-arrestin by GRK2/3.

Moreover, we previously reported that ketamine acted on protein–protein binding in that it inhibited the interaction between one of the GPCR GABA_B_ receptor and GRK4 or GRK5 [38]. As the GRK inhibitor CMPD101 interfered with the ketamine-induced improvement of MOR desensitization caused by fentanyl, the GRK signaling responses could be involved in this ketamine effect. The mechanism of the improvement effects of ketamine appear to be more important in relation to phosphorylated receptors rather than on inactive receptors because neither pretreatment with CMPD101 nor with U0126 in the absence of ketamine improved the desensitization induced by fentanyl or morphine.

After agonists bind to MORs, the receptors are phosphorylated by GRK, and subsequently β-arrestin binds to the phosphorylated sites [39]. Recently, it was shown that there are two β-arrestin binding sites in GPCRs, and the two unique conformations of GPCR-β-arrestin complex elicit different cellular responses. One is the “core” conformation, which induces desensitization of GPCR, and the other is the “tail” conformation, which induces GPCR internalization and resensitization of GPCR [40]. In this study, pretreatment with ketamine with the second administration of fentanyl improved fentanyl-induced MOR desensitization and enhanced β-arrestin recruitment to MORs. These results suggest that ketamine decreases the core conformation via inhibition of β-arrestin binding to MOR or possibly pull β-arrestin out from MOR core sites, resulting in an increase in the numbers of the β-arrestin-bound tail conformation. The tail conformation in MORs continues to activate extracellular-signal-regulated kinase (ERK)1/2, which is activated by MEK1/2, after internalization of MOR, and ketamine is known to activate ERK1/2 in fentanyl desensitization [24]. As our present results showed that β-arrestin activity was increased by ketamine, which is associated with improved desensitization, ERK might also be activated through this process. However, U0126—which suppresses activation of ERK1/2 by inhibiting MEK1/2—failed to suppress the improved effects of ketamine in our study, suggesting that the ERK signal might not be involved in the desensitization process even when ketamine activated ERK1/2.

The benefit of ketamine for opioid tolerance has been reported by several clinical studies. In a randomized controlled trial of spine surgery in patients using opioids for chronic pain, intraoperative ketamine administration at low doses (lower than the anesthetic doses) reduced postoperative opioid tolerance formation and opioid-induced hyperalgesia [41]. In a systematic review on the usefulness of ketamine in patients with cancer, 4 randomized controlled trials and 32 descriptive studies showed that ketamine had the potential to relieve pain in patients who had become inactive or tolerant to opioids [42]. In the present study, 100 nM fentanyl and 10 μM morphine were used in in vitro assays. Some clinical reports have indicated the maximum plasma concentration of fentanyl, morphine and ketamine to be 0.14 μM [43], 77.5 μM [44], and 60–110 μM [45,46], respectively. These data suggest that the doses of the opioids and ketamine used in this study were within the range of clinical concentrations. Accordingly, our present results suggesting that ketamine, at doses within the range of clinical concentrations, improved desensitization induced by fentanyl may in part explain the effectiveness of ketamine against opioid tolerance in the clinical practice.

Cellular and animal studies investigating ketamine’s actions on the effects of opioids, other than analgesia, were not found in the literature. Compared with the sole use of opioids, human studies have reported an increase in adverse events in neurologic and psychiatric events and a decrease in the cardiopulmonary events when ketamine is additionally used with opioids [47]. These results may reflect not only direct effects of ketamine on ORs, but also reductions in opioid dosage and indirect effects via receptors other than ORs. The increase in β-arrestin activity seen in this study when combining opioids and ketamine points to a concerning increase in side effects, such as constipation and respiratory depression, when considering the classical concept of biased agonism [15]. However, it should be noted that the results of this study do not indicate that opioids increase side effects, given that recent studies showed that the β-arrestin pathway in ORs is not directly related to side effects [16].

A limitation of the present study is that we did not directly investigate the changes in the MOR core or tail conformation states induced by ketamine administration. We are presently attempting to establish experiments to observe and calculate the numbers of internalized MORs by ketamine to elucidate the mechanisms induced by β-arrestin signaling. In addition, because we did not conduct in vivo experiments with suitable animal models, further experiments are required to elucidate whether ketamine improves tolerance caused by fentanyl but not morphine.

## 5. Conclusions

Repeated administration of fentanyl or morphine suppressed the consequent MOR responses through MOR desensitization. Administration of ketamine before the second application of fentanyl improved acute desensitization and enhanced β-arrestin recruitment with fentanyl but not with morphine, and the effects of ketamine were suppressed by co-administration of the GRK inhibitor. Our observed responses of ketamine were within the upper limit of clinical concentrations. Our results suggest that ketamine may have improving effects on fentanyl tolerance, in which the conformational changes in GRK and β-arrestin interaction in MOR signaling could be involved and modified by ketamine.

## Figures and Tables

**Figure 1 biomolecules-12-00426-f001:**
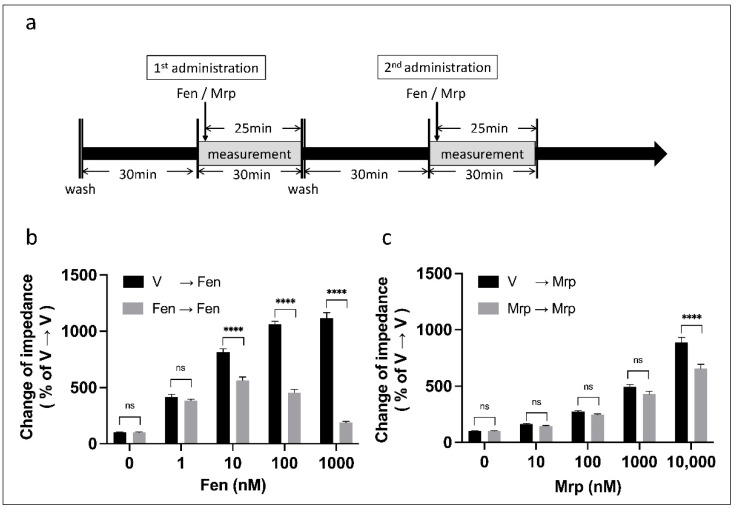
Changes in MOR activity with repeated administration of fentanyl or morphine using the CellKey™ assay. The cells expressing MOR were treated with fentanyl or morphine (first administration) for 25 min. After washing and incubation for 30 min, the same dose of each opioid was administered (second administration) and cellular impedance was measured (**a**). Changes in impedance (ΔZiec) with repeated administration of 1–1000 nM fentanyl (**b**) and 10–10,000 nM morphine (**c**) (two-way ANOVA followed by post hoc Tukey’s test). All data are presented as means ± standard error of mean (SEM) (*n* = 6–12). **** *p* < 0.0001; ns—not significant; V—vehicle; Fen—fentanyl; Mrp—morphine.

**Figure 2 biomolecules-12-00426-f002:**
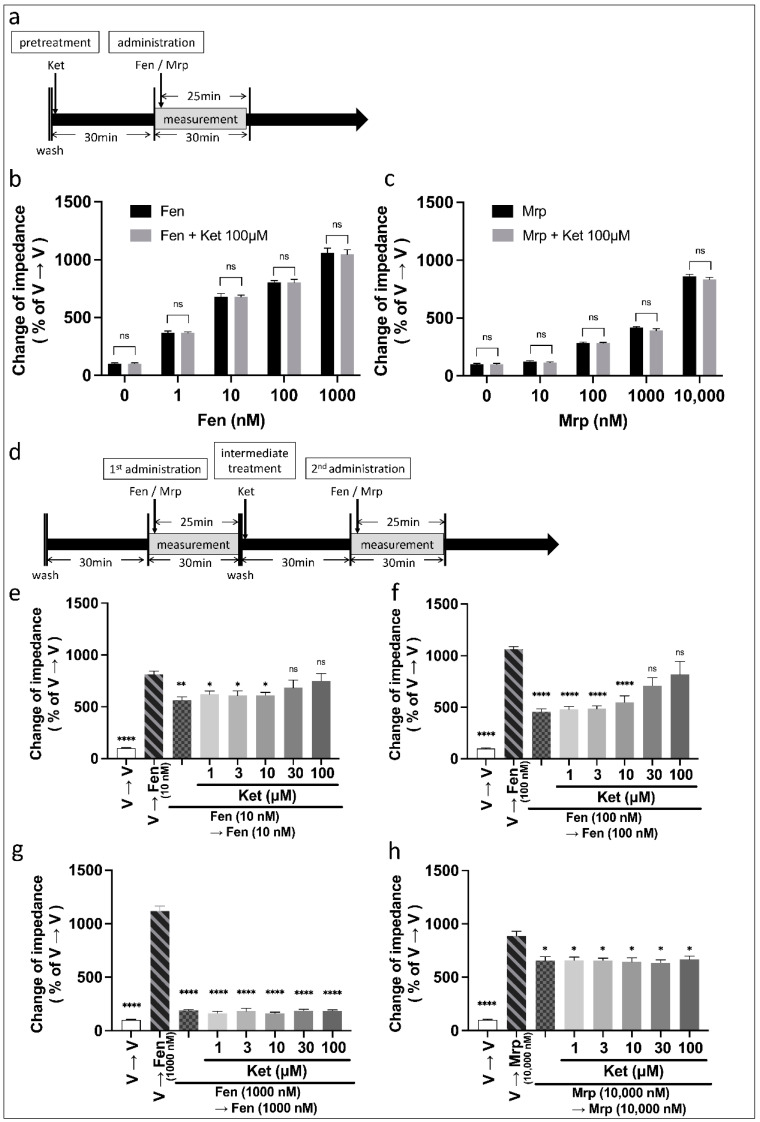
Effects of ketamine on MOR activity induced by single or second administration of fentanyl or morphine in MOR-expressing cells using the CellKey™ assay. Pretreatment with ketamine on single administration of fentanyl or morphine; 100 µM ketamine was pretreated for 30 min before a single administration of fentanyl or morphine (**a**). Effects of pretreatment with ketamine on changes in impedance (ΔZiec) induced by single administration (first administration) of 1–1000 nM fentanyl (**b**) or 10–10,000 nM morphine (**c**) (two-way ANOVA followed by post hoc Tukey’s test). Intermediate treatment with ketamine on repeated administration of fentanyl or morphine; ketamine (1–100 µM) was administered for 30 min before the second administration of fentanyl or morphine (**d**). Effects of intermediate treatment with ketamine on changes in impedance induced by repeated administration of fentanyl at doses of 10 nM (**e**), 100 nM (**f**), 1000 nM (**g**), and 10,000 nM morphine (**h**) (one-way ANOVA followed by post hoc Tukey’s test in comparison to the vehicle to fentanyl or vehicle to morphine groups). All data are presented as means ± SEM (*n* = 6–12). * *p* < 0.05; ** *p* < 0.01; **** *p* < 0.0001; ns—not significant; V—vehicle; Fen—fentanyl; Mrp—morphine; Ket—ketamine.

**Figure 3 biomolecules-12-00426-f003:**
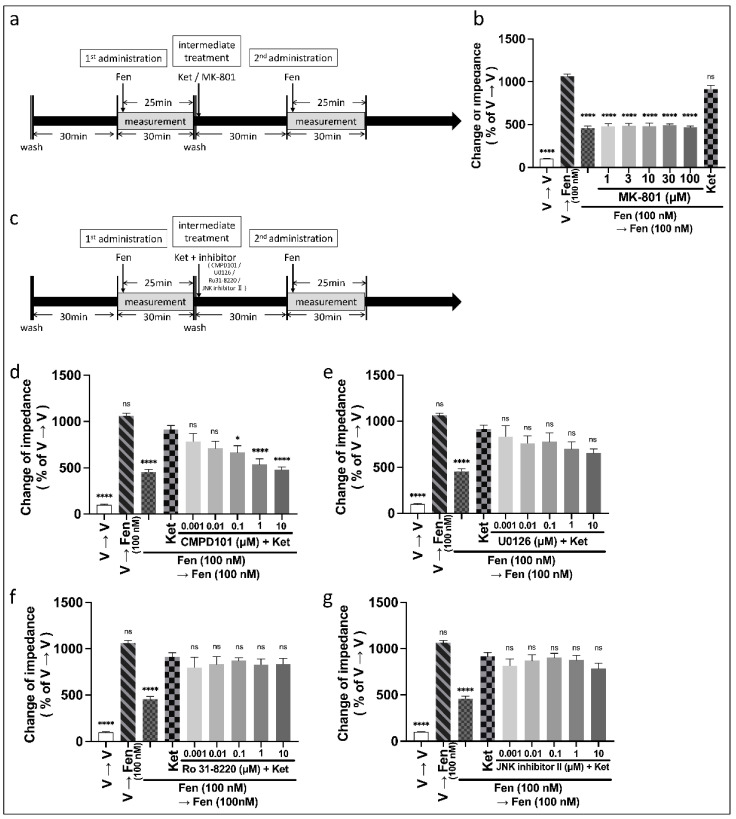
Effects of MK-801 on the decrease in MOR activity caused by repeated administration of fentanyl and intracellular signal inhibitors on ketamine-induced decrease in MOR activity caused by repeated administration of fentanyl in MOR-expressing cells using the CellKey™ assay. MK-801 (1–100 µM) was administered for 30 min before the second administration of fentanyl (**a**). Effects of intermediate treatment of 1–100 µM MK-801 on changes in impedance (Δziec) with repeated administration of 100 nM fentanyl (**b**) (one-way ANOVA followed by post hoc Tukey’s test in comparison to the vehicle to fentanyl group). Each inhibitor was administered concurrently with ketamine (**c**). Effects of impedance in intermediate treatment of CMPD101 (**d**), U0126 (**e**), Ro 31-8220 (**f**), or JNK inhibitor II (**g**) at doses of 0.001–10 µM with 100 µM ketamine on impedance induced by repeated administration of 100 nM fentanyl (one-way ANOVA followed by post hoc Tukey’s test in comparison to the ketamine pretreatment before the second administration of fentanyl group). All data are presented as means ± SEM (*n* = 6–12). * *p* < 0.05; **** *p* < 0.0001; ns—not significant; V—vehicle; Fen—100 nM fentanyl; Ket—100 µM ketamine.

**Figure 4 biomolecules-12-00426-f004:**
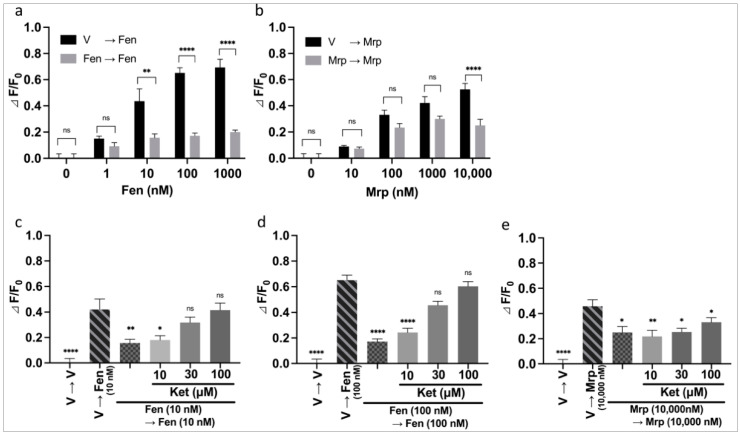
Changes in decrease in intracellular cAMP induced by repeated administration of fentanyl or morphine and effects of intermediate treatment of ketamine on the rescue in intracellular cAMP induced by repeated administration of fentanyl or morphine in MOR-expressing cells using cADDis cAMP assay. Changes in intracellular cAMP with repeated administration at the same dose of 1–1000 nM fentanyl (**a**) and 10–10,000 nM morphine (**b**) (two-way ANOVA followed by post hoc Tukey’s test). Effects of intermediate treatment with 10–100 µM ketamine on the rescue of intracellular cAMP induced by repeated administration of fentanyl at doses of 10 nM (**c**), 100 nM (**d**), and 10,000 nM morphine (**e**) (one-way ANOVA followed by post hoc Tukey’s test in comparison to the vehicle to fentanyl or vehicle to morphine groups). All data are presented as means ± SEM (*n* = 6). * *p* < 0.05; ** *p* < 0.01; **** *p* < 0.0001; ns—not significant; V—vehicle; Fen—fentanyl; Mrp—morphine.

**Figure 5 biomolecules-12-00426-f005:**
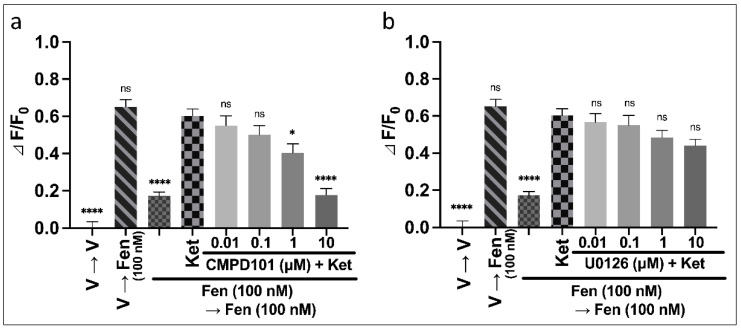
Effects of intracellular signal inhibitors on the rescue of intracellular cAMP induced by repeated administration of fentanyl with ketamine in MOR-expressing cells using cADDis cAMP assay. Effects of 0.01–10 µM CMPD101 (**a**) or U0126 (**b**) on the rescue of intracellular cAMP induced by repeated administration of 100 nM fentanyl with 100 µM ketamine (one-way ANOVA followed by post hoc Tukey’s test in comparison to the ketamine pretreatment before the second administration of fentanyl group). All data are presented as means ± SEM (*n* = 6). * *p* < 0.05; **** *p* < 0.0001; ns—not significant; V—vehicle; Fen—100 nM fentanyl; Ket—100 µM ketamine.

**Figure 6 biomolecules-12-00426-f006:**
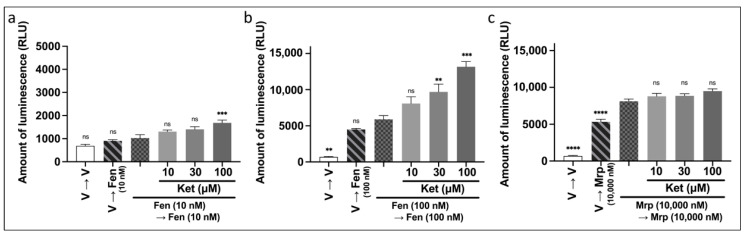
Effects of intermediate treatment with ketamine on changes in β-arrestin recruitment levels caused by repeated administration of fentanyl or morphine in MOR-expressing cells using the PathHunter^®^ eXpress β-arrestin assay. Effects of intermediate treatment with 10–100 µM ketamine on changes in β-arrestin recruitment levels caused by repeated administration of fentanyl at the same doses of 10 nM (**a**), 100 nM (**b**), and 10,000 nM morphine (**c**) (one-way ANOVA followed by post hoc Tukey’s test in comparison to the repeated administration of fentanyl or morphine groups). All data are presented as means ± SEM (*n* = 6). ** *p* < 0.01; *** *p* < 0.001; **** *p* < 0.0001; ns—not significant; V—vehicle; Fen—fentanyl; Mrp—morphine; Ket—ketamine.

**Figure 7 biomolecules-12-00426-f007:**
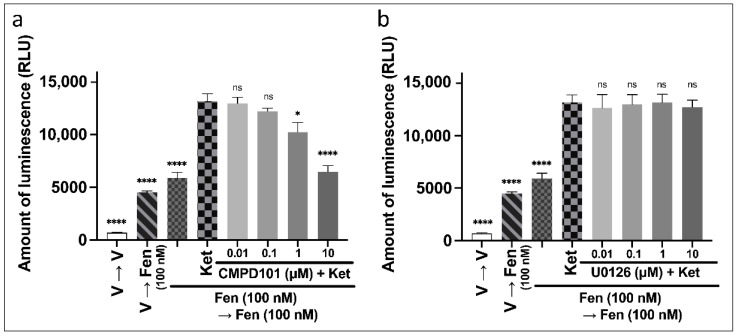
Effects of intracellular signal inhibitors on changes in β-arrestin recruitment levels to MOR induced by repeated administration of fentanyl with ketamine in MOR-expressing cells using the PathHunter^®^ eXpress β-arrestin assay. Effects of 0.01–10 µM of CMPD101 (**a**) or U0126 (**b**) on changes in β-arrestin recruitment to MOR induced by repeated administration of 100 nM fentanyl with 100 µM ketamine (one-way ANOVA followed by post hoc Tukey’s test in comparison to the ketamine pretreatment before the second administration of fentanyl group). All data are presented as SEM (*n* = 6). * *p* < 0.05; **** *p* < 0.0001; ns—not significant; V—vehicle; Fen—100 nM fentanyl; Ket—100 µM ketamine.

## Data Availability

The data that support the findings of this study are available on re-quest from the corresponding author.

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
