# Peer review of "Ketamine Improves Desensitization of µ-Opioid Receptors Induced by Repeated Treatment with Fentanyl but Not with Morphine"

_biomolecules, 2022, doi:10.3390/biom12030426_

Round 1

Reviewer 1 Report

The manuscript "Ketamine Improves Desensitization of µ-Opioid Receptors Induced by Repeated Treatment with Fentanyl but not with Morphine" by Yusuke Mizobuchi et al. addresses the cellular and molecular mechanisms of ketamine-induced improvement in opioid resistance in electrophysiology and fluorescent assays. The article is concise (apart from some sections) and well-written. I have some minor comments regarding the text:

Line 43 "Tolerance is defined as loss of drug potency" I have failed to find such a definition in the literature at a glance. In pharmacology, potency is related indirectly to receptor affinity, and some drug tolerance mechanisms are based on receptor desensitization at a molecular level. However, this does not mean receptor affinity (and potency as well) reduction. Instead, receptor efficacy is affected (the highest agonist effect can be reached). Reference 7 also defines tolerance as efficacy and not potency reduction.

In section 3.2 and throughout the text, I am inclined to recommend the authors reformulate such statements as "the ketamine-induced improvement of suppression of decrease in..._" (lines 306-308) to make them more transparent. Right now, it is incomprehensible, at least for non-native speakers.

In general, the study is performed very well and, in my opinion, can be published after some minor improvements.

Reviewer 3 Report

Which type of fentanyl salt was used? Could the authors verify why they did not go up with concentration of morphine considering that the fentanyl is 100 times more potent than morphine. Please address this point in the paper.
